# Scallop: From Probabilistic Deductive Databases to Scalable Differentiable Reasoning

**Jiani Huang** *
University of Pennsylvania
jianih@seas.upenn.edu

**Ziyang Li** *
University of Pennsylvania
liby99@seas.upenn.edu

**Binghong Chen**
Georgia Institute of Technology
binghong@gatech.edu

**Karan Samel**
Georgia Institute of Technology
ksamel@gatech.edu

**Mayur Naik**
University of Pennsylvania
mhnaik@seas.upenn.edu

**Le Song**
Georgia Institute of Technology
lsong@cc.gatech.edu

**Xujie Si**
McGill University and CIFAR AI Chair, Mila
xsi@cs.mcgill.ca

## Abstract

Deep learning and symbolic reasoning are complementary techniques for an intelligent system. However, principled combinations of these techniques are typically limited in scalability, rendering them ill-suited for real-world applications. We propose Scallop, a system that builds upon probabilistic deductive databases, to bridge this gap. The key insight underlying Scallop is a provenance framework that introduces a tunable parameter to specify the level of reasoning granularity. Scallop thereby i) generalizes exact probabilistic reasoning, ii) asymptotically reduces computational cost, and iii) provides relative accuracy guarantees. On synthetic tasks involving mathematical and logical reasoning, Scallop scales significantly better without sacrificing accuracy compared to DeepProbLog, a principled neural logic programming approach. Scallop also scales to a newly created real-world Visual Question Answering (VQA) benchmark that requires multi-hop reasoning, achieving 84.22% accuracy and outperforming two VQA-tailored models based on Neural Module Networks and transformers by 12.42% and 21.66% respectively.

## 1 Introduction

Integrating deep learning and symbolic reasoning in a principled manner into a single effective system is a fundamental problem in artificial intelligence [10]. Despite great potential in terms of accuracy, interpretability, and generalizability, it is challenging to scale differentiable reasoning in the combined system while preserving the benefits of the neural and symbolic sub-systems [28].

In this paper, we propose Scallop, a systematic and effective framework to address this problem. [2] The key insight underlying Scallop is a principled relaxation of exact probabilistic reasoning via a parameter $k$ that specifies the level of reasoning granularity. We observe that scalability is primarily hindered by reasoning about *all* proofs in computing the probability of each outcome. For a given $k$, Scallop only reasons about the top-$k$ most likely proofs, which asymptotically reduces computational cost while providing formal accuracy guarantees relative to the exact instantiation. Scallop thereby generalizes exact probabilistic reasoning and enables easy exploration of a rich space of tradeoffs.

---

*Jiani Huang and Ziyang Li contributed equally to this work.

[2]The source code of Scallop is available at https://github.com/scallop-lang/scallop-v1.

35th Conference on Neural Information Processing Systems (NeurIPS 2021).

This tradeoff mechanism allows to drastically speed up the stochastic training of the involved neural components without sacrificing generalization ability.

The main technical contribution of Scallop concerns computing the set of top-$k$ proofs associated with each discrete fact *efficiently*, during the evaluation of a logic program, and *correctly*, by maintaining all and only the top-$k$ proofs. Scallop achieves this goal by formulating the problem in the framework of *provenance* for deductive databases [6]. The framework provides the theory and algorithms for tagging discrete facts derived by a logic program with information—in our case the set of top-$k$ proofs. Concretely, Scallop targets Datalog [1], a syntactic subset of Prolog. Although not Turing-complete, Datalog supports recursion and is expressive enough for a wide variety of applications.

Scallop inherits efficient algorithms and optimizations from the databases literature. In contrast, efficiently computing top-$k$ proofs for Prolog is an open problem, to our knowledge. Moreover, the provenance framework enables Scallop to provide correctness guarantees. We leverage the theory of *provenance semirings* [17], which allows us to define how to compute top-$k$ proofs in a compositional manner for each logic operation in Datalog, while ensuring that the computation is correct across arbitrary combinations of these operations. This approach also makes Scallop easy to extend with features such as additional logic operations, probabilistic rules, and foreign functions.

We evaluate Scallop on diverse tasks that involve combining perception with reasoning. On a suite of synthetic tasks that involve mathematical and logical reasoning over hand-written digits, Scallop scales significantly better without sacrificing accuracy compared to DeepProbLog [24], a principled neural logic programming approach. We also create and evaluate on a real-world task called VQAR (Visual Question Answering with Reasoning) which augments the VQA task with an external common-sense knowledge base for multi-hop reasoning. The goal is to answer a programmatic question with the correct subset of objects in a real-world image. Scallop takes 92 hours to finish 15 training epochs with $k = 10$ and takes only 0.3 seconds on average per training sample. In contrast, a difficult training sample can take DeepProbLog over 100 hours to compute, making it infeasible to train on the whole dataset. Scallop's differentiable symbolic reasoning pipeline enables it to achieve 84.22% test accuracy, outperforming two VQA-tailored neural models based on Neural Module Networks and transformers by 12.42% and 21.66% respectively.

In summary, the main contributions of this paper are as follows:

1. We introduce the notion of top-$k$ proofs which generalizes exact probabilistic reasoning, asymptotically reduces computational cost, and provides relative accuracy guarantees.
2. We design and implement a framework, Scallop, which introduces a tunable parameter $k$ and efficiently implements the computation of top-$k$ proofs using provenance in Datalog.
3. We empirically evaluate Scallop on synthetic tasks as well as a real-world task, VQA with multi-hop reasoning, and demonstrate that it significantly outperforms baselines.

## 2   Illustrative Overview

We illustrate our approach using two tasks: a simple task called `sum2` and the real-world VQAR task.

**A Simple Task.** The `sum2` task from [24] concerns classifying sums from pairs of hand-written digits, e.g., $\boxed{3} + \boxed{7} = 10$. As depicted in Figure 1, we specify this task using a neural and a symbolic component, following the style of DeepProbLog [24]. The neural component is a perception model that takes in an image of hand-written digit [20] and classifies it into discrete values $\{0, \ldots, 9\}$. The symbolic component, on the other hand, is a logic program in Datalog for computing the resulting sum. The interface between the neural and symbolic components is a probabilistic database which associates each candidate output of the perception model with a probability. For instance, the fact $0.85 :: \mathsf{d}(\boxed{3}, 3)$ denotes that image $\boxed{3}$ is recognized to be the digit 3 with probability 0.85.

Evaluating the logic program on the probabilistic database yields a weighted boolean formula for each possible result of the sum of two digits, i.e., values in the range $\{0, \ldots, 18\}$. Each *clause* of such a formula represents a different *proof* of the corresponding result. For instance, the bottom left of Figure 1 shows the formula representing all 9 proofs of the ground truth result 10. Each such formula is input to an off-the-shelf weighted model counting (WMC) solver to yield the probability of the corresponding result, e.g., $0.7261 :: \mathsf{sum}(\boxed{3}, \boxed{7}, 10)$.

The scalability of this approach is limited in practice by WMC solving whose complexity is at least #P-hard [31]. We observe that computing only the top-$k$ most likely proofs bounds the size of each formula to $k$ clauses, thereby allowing to trade diminishing amounts of accuracy for large gains in

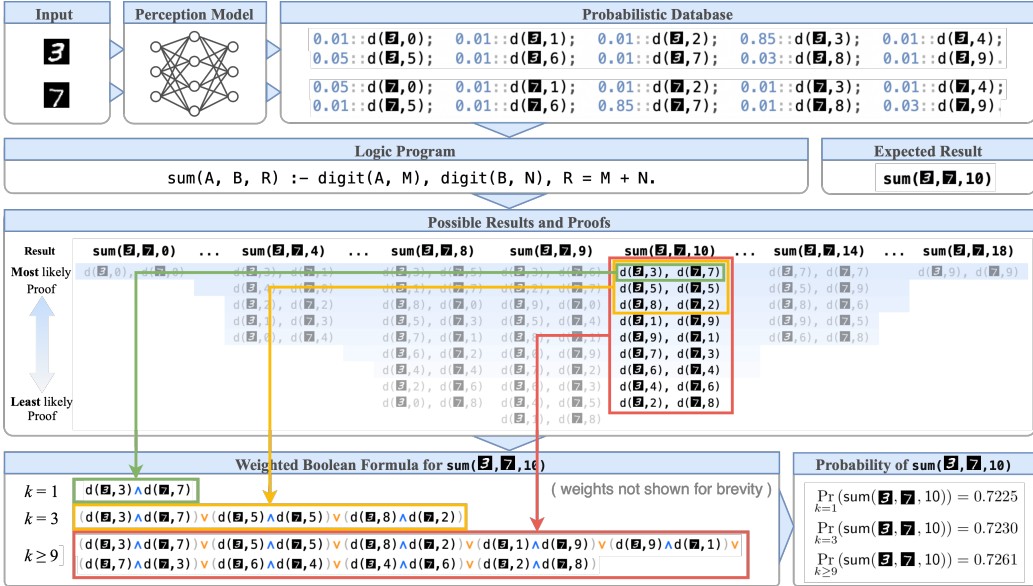

Figure 1: Illustration of our approach on the task 🔢3 + 🔢7 = 10 using different values of parameter $k$.

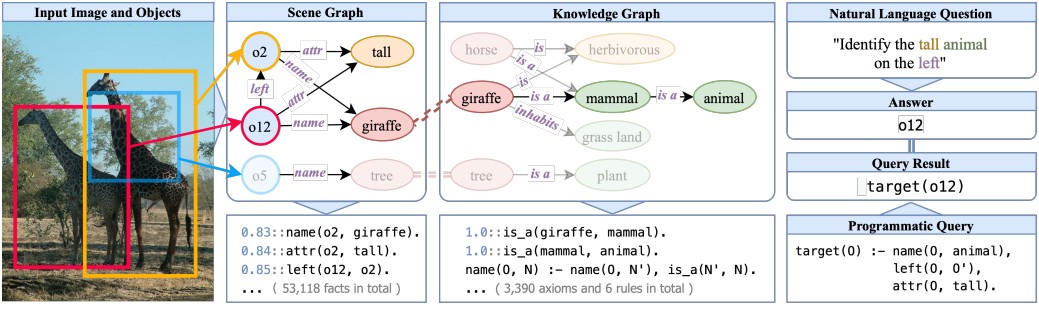

Figure 2: An instance of the VQAR task. The scene graph and knowledge base are shown graphically (above) and in Scallop (below). The question and answer are shown in natural language (above) and in Scallop (below).

scalability. Moreover, stochastic training of the deep perception models itself can tolerate noise in data. As we show later in our experiments, the additional noise introduced by the top-$k$ approximation can be well-compensated for by the stochastic training algorithm.

Scallop embodies this insight by introducing a parameter $k$ which can be task-dependent, and even for a particular task, tuned differently for learning and inference. A higher $k$ leads to slower inference, but accelerates the convergence of learning, especially for complex or sparse feedback; thus, Scallop enables to achieve the best of both worlds by employing a higher $k$ during training, and a lower $k$ thereafter. While Scallop's inference time is under 0.1 second per task for the sum2 task regardless of the choice of $k$, the difference is much more pronounced for the sum3 task of adding three digits: 0.05 seconds for $k = 1$ versus 6.15 seconds for $k = 15$.

**Visual Question Answering.** We next illustrate applying Scallop to a complex real-world task, Visual Question Answering (VQA) [2], which is widely studied in the deep learning literature. The task concerns answering a given question using knowledge from a given image of a scene. Since we are interested in tasks that combine perception with reasoning, we extend the VQA task with *multi-hop reasoning* over an external common-sense knowledge base. The resulting task, which we call *VQAR*, improves upon the VQA task in two important ways: it generalizes the VQA task by allowing questions that require external knowledge, and it allows to precisely control the reasoning complexity through the number of hops needed to answer them. [3] We thereby develop a new dataset consisting of real-world images of scenes and object identification questions that necessitate varying hops of reasoning in a fixed external knowledge base.

---

[3]In contrast, prior works such as the GQA dataset [18] are limited to varying the reasoning complexity in the question alone, which renders the question unweildy.

| (Constant) | $c$ | | |
|---|---|---|---|
| (Variable) | $V$ | | |
| (Term) | $t$ | $V \mid c$ | |
| (Predicate) | $a$ | | |
| (Atom) | $\alpha$ | $a(t_1, \ldots, t_n)$ | |
| (Fact) | $g$ | $a(c_1, \ldots, c_n)$ | $\in \mathcal{G}$ |
| (Input Fact) | $\bar{f}$ | $g$ | $\in \bar{\mathcal{F}}$ |
| (Rule) | $r$ | $\alpha :- \alpha_1, \ldots, \alpha_m$ | $\in \mathcal{R}$ |

| (Probability) | $p$ | | |
|---|---|---|---|
| (Prob. Input Fact) | $f$ | $p :: \bar{f}$ | $\in \mathcal{F}$ |
| (Disjunction) | $j$ | $f_1; \ldots; f_n$ | $\in \mathcal{J}$ |
| (Query) | $\mathcal{Q}$ | $\alpha$ | |
| (Query Result) | $q$ | $g$ | |
| (Program) | $\bar{\mathcal{P}}$ | $(\bar{\mathcal{F}}, \mathcal{R}, \mathcal{Q})$ | |
| (Prob. Program) | $\mathcal{P}$ | $(\mathcal{F}, \mathcal{R}, \mathcal{J}, \mathcal{Q})$ | |

Figure 3: Abstract syntax of probabilistic Datalog programs.

It is natural to express the VQAR task using a combination of neural and symbolic modules akin to the sum2 task. As Figure 2 illustrates, these modules are more complex, reflecting the real-world nature of this task. The neural module is a perception model that takes the object feature vectors (extracted by pre-trained vision models) and outputs a scene graph comprising the predicted name and attribute distributions of each object, and relationships between the objects—all of which are uniformly represented as a probabilistic database. For instance, the tuple $0.83 ::$ name(o12, giraffe) denotes that name of object o12 is classified as giraffe with probability 0.83.

Likewise, the symbolic module uniformly represents both the logic representation of the question and the external knowledge base as a logic program in Datalog.[4] Evaluating the program on the probabilistic database yields the answer, e.g., target(o12). The example in Figure 2 highlights the need for external knowledge: although the question refers to the concept of an "animal" that is missing in the scene graph, Scallop is able to derive the conclusion name(o12, animal) without changing the perception model. The derivation involves two-hop reasoning—two applications of the recursive rule name(O, N) :− name(O, N′), is_a(N′, N) to facts from the scene and knowledge graphs:

$$\frac{\dfrac{\text{name}(\text{o}_{12}, \text{giraffe}) \quad \text{is\_a}(\text{giraffe}, \text{mammal})}{\text{name}(\text{o}_{12}, \text{mammal})} \quad \text{is\_a}(\text{mammal}, \text{animal})}{\text{name}(\text{o}_{12}, \text{animal})}$$

While more sophisticated models can learn the representation of concepts such as animal from a large corpus, relying on such pretrained representation sacrifices the benefits of symbolic reasoning, such as interpretability, data efficiency, and generalization to unseen concepts.

## 3 Background

We recap Datalog, the logic programming language that underlies Scallop, and present its probabilistic extensions that we leverage for inference and training tasks.

**Syntax of Datalog.** As shown in Figure 3, a Datalog program $\bar{\mathcal{P}}$ consists of a set of input facts $\bar{\mathcal{F}}$, a set of rules $\mathcal{R}$, and a query $\mathcal{Q}$. The building block is an atom $a(t_1, \ldots, t_n)$ which consists of an $n$-ary predicate $a$ and a list of terms $t_1, \ldots, t_n$ as arguments. A fact $g$ is an atom which all the argument terms are constants; it may be an input fact (EDB) or a derived fact (IDB). Datalog rules are of the form $\alpha :− \alpha_1, \ldots, \alpha_m$, meaning that atom $\alpha$ in the head is true if all atoms $\alpha_i$ in the body are true. Multiple rules sharing a single head predicate denote disjunction (or union).

**Semantics of Datalog.** Datalog programs can be executed using a bottom-up evaluation strategy. Starting from the input facts $\bar{\mathcal{F}}$, we repeatedly apply the rules $\mathcal{R}$ in any order to derive new facts until a fixed point is reached. Upon completion, we obtain all the output facts $q$ of the query $\mathcal{Q}$. For example, with $\bar{\mathcal{F}} = \{\text{left}(\text{o}_1, \text{o}_2), \text{below}(\text{o}_2, \text{o}_3)\}$ and $\mathcal{Q} = \text{left}(\text{o}_1, \text{O})$, the execution of program $(\bar{\mathcal{F}}, \emptyset, \mathcal{Q})$ produces $\{\text{left}(\text{o}_1, \text{o}_2)\}$. We denote the execution result as $\texttt{Exec}(\bar{\mathcal{P}}) = \{q_i\}_{i=1}^{n}$.

**Probabilistic Extensions.** To handle uncertain data, we introduce two probabilistic extensions to Datalog, which are inspired by pD [15] and ProbLog [11]. First, we specify probabilistic input facts $f$ by associating a probability $p$ with $\bar{f}$, declaring that $\Pr(f) = p$. Deterministic input facts have probability 1.0. Secondly, we allow disjunctions $\mathcal{J}$ among probabilistic input facts, denoted by $f_1; \ldots; f_m$. For example, the disjunction

$$0.01 :: \text{digit}(\boxed{3}, 0); \ldots; 0.82 :: \text{digit}(\boxed{3}, 3); \ldots; 0.06 :: \text{digit}(\boxed{3}, 9).$$

states that the digit $\boxed{3}$ is recognized to be 0 to 9 with their respective probabilities, but cannot be more than one simultaneously. $\mathcal{F}$ and $\mathcal{J}$ form a *probabilistic database*. By combining the $\mathcal{F}$, $\mathcal{J}$ with $\mathcal{R}$ and $\mathcal{Q}$, we obtain a probabilistic Datalog program $\mathcal{P}$.

---

[4]We presume that the input question is in programmatic form because existing models for semantic parsing achieve high accuracy in translating from natural language text to programmatic form [5].

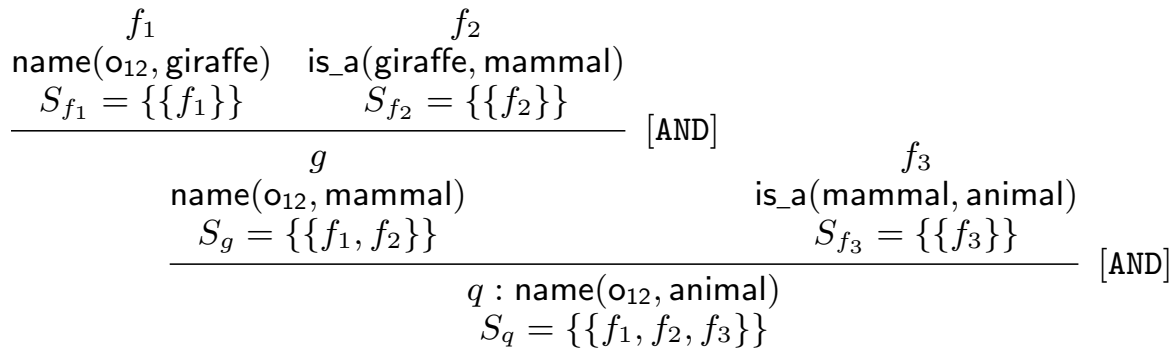

Figure 4: Proof constr. with conjunction.

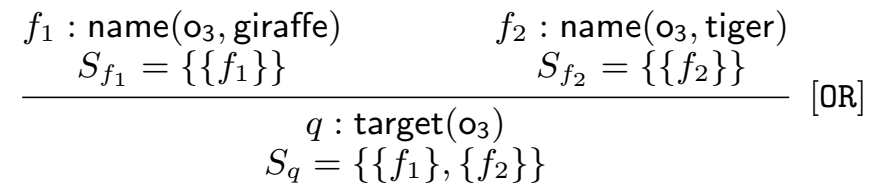

Figure 5: Proof constr. with disjunction.

**Probability Calculation.** Unlike discrete Datalog, which provides definite answers to queries, we wish to compute the *success probability* of each query result $q$: $\texttt{Exec}(\mathcal{P}) = \{(q_i, \Pr(q_i))\}_{i=1}^n$. To compute success probabilities, we first define a *proof* of any fact $g$ as a minimal set of (probabilistic) input facts $f$ that can derive $g$. We denote a proof as $F \in \mathbb{P}(\mathcal{F})$ where $\mathbb{P}$ denotes power set. Since a fact $g$ may be explained by multiple proofs, we use $S_g$ to denote the complete set of proofs of $g$. Given the set of proofs $S_q$ for a query result $q$, the success probability $\Pr(q)$ is simply the likelihood of $S_q$, denoted $\Pr(S_q)$, which can be computed using *Weighted Model Counting* (WMC) [19].

# 4 Framework

Scallop aims to solve the following two problems:

1. **Inference** (Section 4.1): Given a probabilistic Datalog program $\mathcal{P} = (\mathcal{F}, \mathcal{R}, \mathcal{J}, \mathcal{Q})$, efficiently compute each query result $q_i$ with its set of proofs $S_{q_i}$.
2. **Learning** (Section 4.2): Given a neural symbolic reasoning dataset $\mathcal{D}$ and a loss function $\mathcal{L}$, learn a perception model $\mathcal{M}_\theta$ which, for each $(x, y) \in \mathcal{D}$, transforms $x$ into a probabilistic database captured by Datalog program $\mathcal{P}_\theta^x$. We aim to minimize the following objective: $J(\theta) = \frac{1}{|\mathcal{D}|} \sum_{(x,y)\in\mathcal{D}} \mathcal{L}\big(\texttt{Exec}(\mathcal{P}_\theta^x), y\big)$.

## 4.1 Inference

**Proof Construction.** The goal of our proof construction is to construct the set of proofs $S_q$ for every query result $q$. We can efficiently compute $S_q$ during the bottom-up execution of the Datalog program. We initially tag each input fact $f \in \mathcal{F}$ with $S_f = \{\{f\}\}$ and propagate proofs during execution from known facts to newly derived facts.

We illustrate proof propagation during conjunction in Figure 4. When $g$ is derived from a conjunction on $f_1$ and $f_2$, we combine the sets of proofs $S_{f_1}$ and $S_{f_2}$ to produce $S_g$. The resulting $S_g$ contains a single proof $\{f_1, f_2\}$, as both $f_1$ and $f_2$ must be true for $g$ to be true. More formally, we define a binary operation $\otimes$ corresponding to conjunction. Given two sets of proofs $S_1$ and $S_2$, we have

$$S_1 \otimes S_2 = \{F \mid F = F_1 \cup F_2, (F_1, F_2) \in S_1 \times S_2,\ F \text{ contains no disjunction conflict}\}. \quad (1)$$

We next illustrate proof propagation during disjunction in Figure 5. Consider a VQAR instance in which the query concerns identifying a target object that is either a giraffe or a tiger. $S_q$ contains two separate proofs, one containing only $f_1$ and the other containing only $f_2$, as each can individually explain $q$. We thereby define a binary operation $\oplus$ corresponding to disjunction, as set union:

$$S_1 \oplus S_2 = S_1 \cup S_2. \quad (2)$$

Equipped with $\oplus$ and $\otimes$, we can show that the collection of sets of proofs $\mathcal{S} = \mathbb{P}(\mathbb{P}(\mathcal{F}))$ forms a semiring, which we call the *proof semiring*. Following [17], every derivable fact $g$ can be annotated with a corresponding algebraic formula representing the bottom-up construction of $S_g$. Since the proof semiring is both commutative and distributive, we show in Appendix A.1 that $S_q = \bigoplus_{F \text{ derives } q} \Big(\bigotimes_{f\in F} S_f\Big)$.

However, the complexity of $S_q$ renders the computation infeasible. In principle, we have $|S_q| = \mathcal{O}(2^{|\mathcal{F}|})$, showing that $|S_q|$ grows exponentially with the amount of input facts. The actual version of our example shown in Figure 2 generates 2,619 proofs in total for all query results, and takes 14 minutes to execute. This scalability issue is further exacerbated when the system is used in a learning setting, where we need to execute millions of such programs.

**Top-$k$ Proof Construction.** The probabilistic nature of our problem setting opens up room for approximation. A key observation is that, when the inference system is used in a learning setting, the probability of a ground truth fact should significantly outweigh other facts, forming a skewed distribution. We can exploit this property by only including the "most likely" proofs in $S_q$, with the likelihood of a proof $F$ defined by $\Pr(F) = \prod_{f\in F} \Pr(f)$.

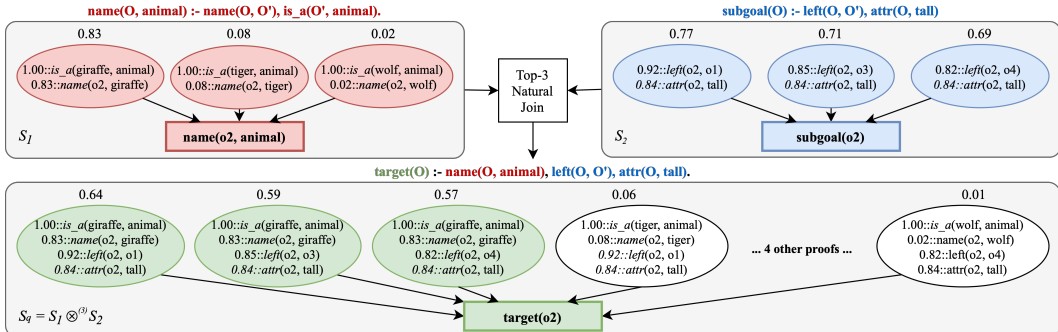

Figure 6: Illustration of top-$k$ natural join using $k = 3$. Each ellipse represents a proof of the fact shown in the box. Given the top 3 proofs for each of "name($o_2$, animal)" and "subgoal($o_2$)", we wish to derive the top 3 proofs for their conjunction, "target($o_2$)". The join yields 9 possible proofs. After computing the likelihood for each of the 9 proofs, we keep the top 3 most likely ones (green ellipses) and discard the rest (white ellipses).

We thereby introduce a *top-$k$ proof inference* algorithm. With a user-specified hyper-parameter $k \geq 1$, we perform top-$k$ filtering at each step of the proof construction. We define two new operations, $\otimes^{(k)}$ for conjunction, and $\oplus^{(k)}$ for disjunction:

$$S_1 \otimes^{(k)} S_2 = \texttt{Top}_k(S_1 \otimes S_2), \qquad S_1 \oplus^{(k)} S_2 = \texttt{Top}_k(S_1 \oplus S_2). \qquad (3)$$

Intuitively, whenever $\otimes$ or $\oplus$ is performed, we rank proofs by their likelihood and preserve only the top-$k$ proofs. This allows us to discard the vast majority of proofs and thus make inference tractable. An example run-through of *top-3 natural join* ($\otimes^{(3)}$) is depicted in Figure 6, where we perform a normal $\otimes$ operation followed by a top-3 filtering.

As before, we construct a *top-$k$ proof semiring* (Appendix A.2), with which we can express the resulting approximated *beam of proofs* $\tilde{S}_q = \bigoplus^{(k)}_{F \text{ derives } q} \left( \bigotimes^{(k)}_{f \in F} S_f \right)$. Note that the size of $\tilde{S}_q$ is bounded by $k$, $|\tilde{S}_q| = \mathcal{O}(k)$, reducing the exponential complexity of exact inference to a near constant one. As a comparison point, with top-3 proof inference, the full example shown in Figure 2 only generates 39 proofs, taking only 0.5 seconds to execute. Formally, our approximation of the *success probability* of a given query result $q$ can be written as $\Pr(q) = \Pr(S_q) \approx \Pr(\tilde{S}_q)$.

**Discussion.** We present some desirable properties of our top-$k$ inference algorithm. The approximation error bound is given by $|\Pr(S_q) - \Pr(\tilde{S}_q)| \leq \sum_{F \in S_q \setminus \tilde{S}_q} \Pr(F)$, and we can tune $k$ to control the trade-off between scalability and accuracy. Furthermore, if no disjunctions are specified ($\mathcal{J} = \emptyset$), then we have $\tilde{S}_q = \texttt{Top}_k(S_q)$, that is, the beam of proofs $\tilde{S}_q$ contains the global top-$k$ proofs. The theorems and proofs are provided in Appendix A.3.

We also note that our top-$k$ inference algorithm is reminiscent of beam search. Both methods are iterative and explore only the top-$k$ elements at each step. However, there are two major differences that distinguish us from beam search. First, while beam search is heuristic, our algorithm is backed by Datalog semantics and the provenance semirings framework for its correctness. We also present formal guarantees on its approximation error bound. Secondly, our algorithm operates over the beam of proofs $\tilde{S}_q$ for each derived fact $q$, while beam search is usually performed to search for an output.

### 4.2 Learning

At a high level, we want to train a perception model $\mathcal{M}_\theta$ that takes in an input $x$ and produces a probabilistic database $(\mathcal{F}, \mathcal{J})$, captured by program $\mathcal{P}$, such that after execution, can derive the ground truth $y$ as the output. Note that the probability of the input facts in the probabilistic database is generated by the perception model $\mathcal{M}_\theta$. Therefore each input probability $p_i = \Pr(f_i)$ is also associated with their gradients $\nabla_{\Pr(f_i)}$ with respect to the model parameters $\theta$.

To back-propagate the gradients through the inference process, similar to DeepProbLog [24], Scallop adopts a *gradient semiring* augmented WMC procedure, for which we use *Sentential Decision Diagram* (SDD) [9]. The beam of proofs $\tilde{S}_q$ will be transformed into a weighted Conjunctive Normal Form (CNF) formula, where for each variable, $f_i$, we attach the dual number $(\Pr(f_i), \nabla_{\Pr(f_i)})$ as its weight. As a result, the associated differentiable probability of each query result $q_i$ will be $(\Pr(q_i), \nabla_{\Pr(q_i)})$, as computed by WMC. With everything above, we define the execution of our

| Task | Goal Predicate | #Out | Max #Proofs | Scallop | | | | DPL |
|------|----------------|------|-------------|---------|---------|---------|---------|-----|
| | | | | $k=1$ | $k=3$ | $k=5$ | $k=10$ | |
| **T1** | sum2(**3**, **7**, 10) | 19 | 10 | **97.46%** | 96.90% | 96.67% | 96.29% | 96.82% |
| **T2** | sum3(**3**, **7**, **5**, 15) | 28 | 75 | 95.31% | 95.43% | 95.76% | **95.76%** | 95.56% |
| **T3** | sum4(**3**, **7**, **5**, **2**, 17) | 37 | 670 | 47.11% | **95.47%** | 95.31% | 95.07% | – |
| **T4** | sort2(**3**, **7**, 0, 1) | 2 | 55 | 80.43% | 91.55% | 91.75% | 95.49% | **98.04%** |
| **T5** | sort3(**7**, **2**, **3**, 1, 2, 0) | 6 | 220 | 70.34% | 93.20% | 96.15% | **97.09%** | 95.50% |
| **T6** | sort4(**7**, **3**, **5**, **2**, 3, 1, 2, 0) | 24 | 715 | 68.67% | 87.90% | **92.02%** | 91.87% | 89.96% |

Table 1: Testing accuracy of Scallop and DeepProbLog (DPL) on a suite of 6 synthetic tasks. All numbers except $k=1$ have a standard deviation of $< 2\%$.

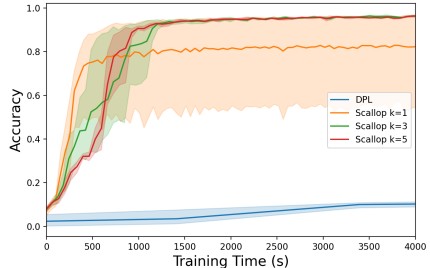

Figure 7: Training runtime (in seconds) vs. validation accuracy for task **T2** (sum3).

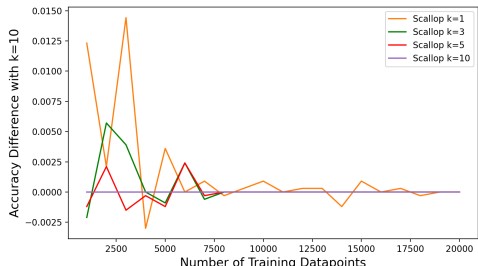

Figure 8: Difference in accuracy of varying $k_{\text{test}}$ compared to $k_{\text{test}} = 10$ for task **T2** (sum3).

probabilistic Datalog program as

$$\hat{y} = \texttt{Exec}(\mathcal{P}) = \{(q_i, (\Pr(q_i), \nabla_{\Pr(q_i)}))\}_{i=1}^{n}. \tag{4}$$

The results of the execution $\hat{y}$, along with the ground truth $y$ is passed to the given loss function $\mathcal{L}$. Lastly, the loss is back-propagated to update $\theta$, the parameters of the perception model $\mathcal{M}_\theta$.

For example, the ground truth label $y$ for the task sum(**3**, **7**, R) is a binary vector of dimension 19, conceptually representing the set:

$$\{0.0 :: \text{sum}(\textbf{3}, \textbf{7}, 0), \ldots, 1.0 :: \text{sum}(\textbf{3}, \textbf{7}, 10), \ldots, 0.0 :: \text{sum}(\textbf{3}, \textbf{7}, 18)\}.$$

and the predicted $\hat{y}$ is a set of the 19 results associated with their predicted probabilities, represented as a probability vector of dimension 19. In our experimental setup, we apply the binary cross entropy loss function on the two vectors. In practice, however, the loss function is fully customizable.

## 5 Evaluation

We evaluate Scallop on a suite of synthetic tasks and VQAR. All experiments are conducted on a machine with two 20-core Intel Xeon CPUs, four GeForce RTX 2080 Ti GPUs, and 768 GB RAM. Experimental details such as hyperparameter selection and dataset splits are provided in Appendix C, and implementation details of the Scallop framework are explained in Appendix D.

### 5.1 Synthetic Tasks

We extend the synthetic tasks from DeepProbLog (DPL) to demonstrate that (1) Scallop is much more scalable, (2) Scallop does not sacrifice accuracy, and (3) how different levels of reasoning granularity during training and testing phases can affect model performance.

Table 1 shows 6 synthetic tasks and their corresponding sample goal predicates. Each task takes as input multiple MNIST [20] images and requires performing simple arithmetic (**T1**-**T3**) or sorting (**T4**-**T6**) over digits depicted in the given images. The difficulty of each task is reflected by third and fourth columns, which show the size of the output space and the maximum number of proofs per output, respectively. Our goal is to train a digit classifier end-to-end with the combined perception + reasoning pipeline. We elaborate on individual tasks further in Appendix E.

**Accuracy.** We show accuracy comparison with DPL in Table 1. All models are trained under the same learning setting. Scallop is able to achieve on par accuracy as DPL, despite using far fewer proofs. It also shows that in general, larger $k$ implies better accuracy. Note that we are unable to collect result for DPL on **T3**, as DPL takes 24 hours only to complete 100 out of the 15,000 training samples. In contrast, Scallop with $k=3$ finishes 5 epochs (75,000 training samples) within 4 hours.

| Test Dataset | LXMERT | NMNs | Scallop |
|---|---|---|---|
| 1000 C2 | 66.75% | 79.32% | **85.17%** |
| 1000 C3 | 61.69% | 61.98% | **82.82%** |
| 1000 C4 | 63.82% | 71.17% | **83.25%** |
| 1000 C5 | 64.05% | 74.62% | **85.53%** |
| 1000 C6 | 56.51% | 72.04% | **84.30%** |
| 5000 $C_{all}$ | 62.56% | 71.80% | **84.22%** |

Table 2: Testing accuracy (in Recall@5) of Scallop, NMNs, and LXMERT on VQAR dataset.

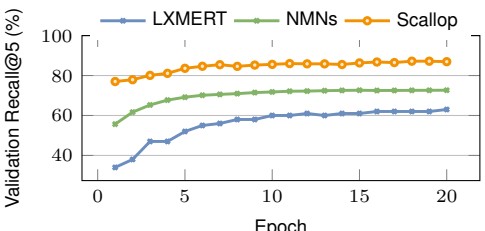

Figure 9: Results of training on 50K $C_{all}$ tasks and testing on 5000 tasks of different clause lengths.

**Runtime vs. Accuracy.** We next evaluate the tradeoff between the training runtime vs. testing accuracy in Scallop. Figure 7 shows the results for the sum3 task. With $k = 1$, Scallop learns the fastest in the beginning, but it has high variance and potential of failing to converge to an optimal solution. On the other hand, with $k = 5$, it has much less variance and converges the fastest despite being slower in the beginning. We compare with DPL trained under the same setting. It achieves the same accuracy (95.56%) at the end of the 3rd epoch, but due to its long runtime (14 hours), we omit showing the whole curve in this figure.

**Decoupling Reasoning Granularity.** Scallop enables using different $k$ during training and testing phases. The key idea is that a larger $k$ will help faster convergence in training, whereas a smaller $k$ suffices during testing since less probable proofs have minimal impact on the reasoning result. In Figure 8, we fix a $k_{train} = 10$ on the sum3 task. Taking accuracy with $k_{test} = 10$ as a baseline, we compute the difference in testing accuracy on $k_{test} \in \{1, 3, 5\}$. The figure shows that as the training progresses, the difference converges to $0\%$. This suggests we can tune $k_{train}$ and $k_{test}$ individually for better training as well as faster test time inference.

## 5.2 Visual Question Answering

We next evaluate Scallop on the VQAR task described in Section 2. Besides DPL, we compare with two neural methods: Neural Module Network (NMN) and LXMERT, a transformer based approach.

**Dataset.** The VQAR dataset contains (a) 80,178 images, (b) object feature vectors + bounding boxes, (c) scene graphs with 500 object names, 609 attributes, and 229 relationships, (d) a shared knowledge graph with 6 rules and 3K knowledge triplets, and (e) 4M programmatic queries and answer pairs. The images and scene graphs are from the GQA [18] dataset and the knowledge graph is from the CRIC [16] dataset. The object feature vectors and bounding boxes are then obtained by passing the images through pre-trained fixed-weight Mask RCNN and ResNet models. Using random walk on combined scene graph and external knowledge graph, we generate object identification questions in the form of programmatic queries. We further categorize these queries into different levels of difficulty by the number of occurring clauses from C2 to C6, where C2 is the simplest and C6 is the hardest. For each image, we generate 10 different question and answer pairs for each clause length 2 to 6, to obtain 4 million data points in total. We split the images randomly into training (60%) validation (10%), and testing (30%) sets. Further details of this dataset are provided in Appendix B.

We formulate VQAR as a multi-label classification task. For each datapoint $(x, y)$ in our VQAR dataset, the input $x$ consists of (a) the *entire* knowledge graph $KG$, (b) a programmatic query, and (c) the object feature vectors and bounding boxes. The ground truth $y$ is the set of objects that the given programmatic query identifies. All of our evaluated models share this same set of input and output (except LXMERT, which takes in natural language questions instead of programmatic queries). The accuracy is measured by Recall@5.

**Setup of Scallop.** We use a perception module consisting of three MLP-classifiers, $\mathcal{M}_\theta = (\mathcal{M}_\theta^n, \mathcal{M}_\theta^a, \mathcal{M}_\theta^r)$, which predict names, attributes, and relations respectively. All predictions are transformed into probabilistic facts in a database. The outputs of $\mathcal{M}_\theta^n$ form disjunctions because each object has only one name. With $KG$ as part of the probabilistic database, we perform Datalog execution on the given programmatic query to obtain the set of identified objects. Note that the *entire* knowledge graph is used in every Datalog execution. We use binary cross entropy as our loss function to compare the predicted set of objects and the ground truth set. The goal is to train the three classifiers in Scallop end-to-end, and identify the correct objects according to the question.

**Baseline 1: DeepProbLog.** It is prohibitively slow to train with DPL from scratch—a regular training sample from C6 can take DPL more than 100 hours to run. Therefore, instead of training

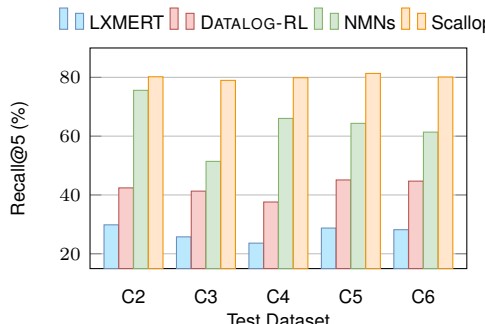
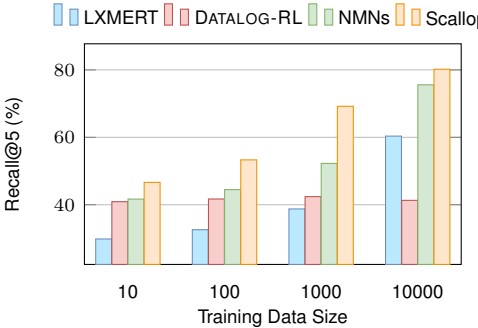

Figure 10: Generalizability to harder questions when trained on 10K C2.

Figure 11: Data efficiency given training data size from 10 to 10,000 C2.

with DPL, we use the perception model $\mathcal{M}_\theta$ trained with Scallop to test DPL's inference capability. With 10 seconds timeout, DPL times out on 68.66% of the testing samples, while Scallop finishes all with an average running time under 0.3 seconds per sample.

**Baseline 2: Neural Module Network.** We compare against RVC [16], a Neural Module Network approach for VQA with external common-sense knowledge. This method first pretrains a TransE embedding [3] for the knowledge graph. Then, to mimic the reasoning process, it trains a set of neural modules that perform knowledge retrieval, scene graph traversal, and logical operations. The modules are assembled according to the programmatic query and can leverage object-based features.

**Baseline 3: LXMERT.** We also compare to LXMERT [33], a recent transformer based approach that emphasizes its transfer learning ability. LXMERT takes in a natural language question corresponding to the given programmatic query. Similar to other baselines, the object features and bounding boxes are taken as input. Since this model cannot explicitly use a knowledge base, we leverage the implicit relations learned through pre-training over a variety of image-language tasks: MS COCO [22], Visual Genome [2], and GQA [18]. Finally, we fine-tune LXMERT on our VQAR training samples.

**Ablation Study: Datalog Reinforcement Learning (DATALOG-RL).** In this study, we remove the differentiability in Scallop's learning pipeline. Instead, we sample a discrete scene graph, run it through the standard Datalog execution, and use the overlap in predicted objects as a reward to estimate the gradient using REINFORCE [35]. This method does not scale with the training dataset of 50K tasks, so we only perform the generalizability experiments (Figure 10).

**Results.** Table 2 and Figure 9 compares the performance of Scallop, NMNs, and LXMERT based on 50K training tasks. Scallop significantly outperforms both in terms of accuracy and data efficiency. Figure 10 shows that Scallop generalizes to answer more difficult questions (1K from each of C2-C6) even when trained on only the easiest ones (10K C2). Figure 11, on the other hand, shows the testing accuracy (on 1K C2) when trained on varying dataset sizes (10, 100, 1000, and 10,000 C2). We observe that Scallop has the best data efficiency. Finally, with DATALOG-RL we observe that the addition of differentiable reasoning is crucial to Scallop's learning performance.

## 6 Discussion and Limitations

**Top-k hyper-parameter selection.** The hyper-parameter k is much easier to tune than a traditional one due to its deterministic behavior. At training time, a lower k means faster inference time, and a higher k means higher inference accuracy. Note that sometimes a higher k may lead to faster convergence than a lower k. That is because the higher k means more proofs will be considered during the weighted model counting process. Subsequently, more gradients will be back-propagated to the source, resulting in faster convergence of learning. At testing time, k merely affects whether we consider certain low probability proofs. Therefore it will likely have less impact on the prediction result. For both the synthetic tasks and the VQAR task we performed, we found k=5 to be a suitable default value that balances accuracy and training cost. In practice, the user may start with k = 5, then, increase or decrease this value to achieve higher accuracy or lower training cost, respectively.

**Scaling to large knowledge bases.** In the real world, incorporating a larger knowledge base is helpful to avoid failures due to incomplete knowledge base and vocabulary. We estimate the efficiency of Scallop with regards the sizes of the knowledge base. For the knowledge base with 3K triplets,

it takes Scallop 0.2 seconds on average to process one query. When we use a subset of the ConceptNet knowledge base comprising 250K triplets with the same Scallop implementation, the time consumption per query increased to 2 seconds. Although Scallop runs fast with non-trivial-sized knowledge bases, to incorporate an even larger knowledge base such as the entire ConceptNet (34M) or WikiData (94M) will require system-level optimizations and is beyond the scope of this paper.

**Programming interface.** The Scallop framework provides a generic interface for performing differentiable logical inference. The input to our interface is (1) a probabilistic relational database $(\mathcal{F}, \mathcal{J})$ consisting of tuples with associated probabilities (with gradients) that encodes the output of the neural components, and (2) a set of Datalog rules $\mathcal{R}$ that specifies the logic reasoning components. The output is the probabilistic query results, which can be either used to calculate the loss directly or as the input to subsequent neural components. The Scallop framework is able to capture a variety of machine learning tasks such as the examples shown in Appendix G.

**Natural language questions.** In our VQAR task, the query is given in its programmatic form. However, in the generic setup of the visual question and answering (VQA) task, a question is usually provided in its natural language form. To convert a natural language question into its programmatic form, the user may need to train a separate model for semantic parsing. Automatically generating such a program with end-to-end reasoning using program synthesis, semantic parsing, or inductive logic programming techniques is an interesting but orthogonal future direction.

## 7 Related Work

**Neural symbolic methods.** Neural symbolic methodology aims to disentangle low-level perception from high-level reasoning systematically. Generically speaking, there are three classes of the neural symbolic method. (1) Logic regularization term. Whenever the network fails to obey the logic constraint, it will receive a penalty [32, 36]. (2) Soft logic program execution. The primitive operations in a logic program are mapped to differentiable mathematical operations or neural components [14, 29]. (3) Proof-guided probability calculation. Approaches like exact probability calculation and abductive reasoning first execute the logic program and then map the generated proof constructs into differentiable expressions [8, 21, 24].

Using logic constraints as regularization terms can scale, but does not guarantee the reasoning correctness. Substituting logic reasoning steps by differentiable components fails to preserve the original semantics of logic reasoning. Exact probability calculation, on the other hand, maintains the purity of the logic reasoning pipeline, but has significant scalability limitation. Most application-specific neural symbolic approaches fall in categories (1) and (2) due to their high-efficiency demand.

**Scaling reasoning algorithms.** Other neural symbolic methods have explored optimization strategies for their reasoning algorithms. Neural Theorem Prover (NTP) [30] considers all reasoning paths in the inference procedure. Due to its high computation cost, subsequent works focus on improving its scalability. For instance, Greedy NTP [26] keeps a beam of proof states using nearest neighbor search. Another notable example is Conditional Theorem Prover [27] which applies soft proof selection by training a neural network to select the rules, deriving proofs individually.

**Forward and backward chaining.** Methods such as Scallop and TensorLog [7] apply *forward chaining*, a reasoning method that derives conclusion from known facts and rules. In particular, Scallop employs Datalog and a probabilistic deductive database to derive all possible query results. This is as opposed to *backward chaining* methods, such as (Deep)ProbLog and NTP, which start from the goals and work backwards to determine if any data supports the goal.

## 8 Conclusion and Future Work

We proposed Scallop, a framework for scaling differentiable reasoning based on Datalog, motivated by real-world applications that necessitate combining perception and reasoning. The key idea underlying Scallop is to relax exact probablistic reasoning via a tunable parameter that specifies the level of reasoning granularity. We demonstrated the effectiveness of Scallop on diverse tasks including a newly created Visual Question Answering benchmark that requires multi-hop reasoning. In future, we plan to develop expressive extensions to Scallop, target more challenging neuro-symbolic applications, and optimize the end-to-end pipeline on modern hardware.

**Acknowledgements.** We thank our anonymous reviewers for valuable feedback. This research was supported by grants from ONR (#N00014-18-1-2021), NSF (#2107429 and #1836936), and the Canada CIFAR AI Chair Program.

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
