# OpenReview forum: "Scallop: From Probabilistic Deductive Databases to Scalable Differentiable Reasoning"
_NeurIPS.cc/2021/Conference — NeurIPS 2021 Poster_

### Official Review · Reviewer_2nJQ · 2021-07-15

**Rating:** 5
**Confidence:** 4

**Summary:**

The paper proposes Scallop: a system for differentiable reasoning on probabilistic deductive databases. Scallop improves upon prior works in terms of scalability. Its core idea is to consider only top-k proofs rather than all proofs during deductive inferences. The method can be implemented efficiently leveraging provenance for deductive databases. Experiments on a synthetic addition task and a real-world VQA task show that Scallop outperforms baselines in speed and accuracy.

**Limitations And Societal Impact:**

## Technical Limitations

See my comments in "Questions and Weaknesses"

## Societal Impact

I do not see any potential negative societal impact.

**Main Review:**

## Strengths

+ Having to consider all proofs is a speed bottleneck for neuro-symbolic reasoning. This paper addresses it by keeping only the top-k proofs in each reasoning step. It is an intuitive and straightforward idea since many proofs have marginal effects on the score of the final answer.

+ The authors propose an elegant implementation for selecting top-k proofs using provenance—a mechanism developed in the deductive database community for attaching additional information to the derived facts. It enables using highly optimized off-the-shelf datalog solvers, without having to implement the inference algorithms from scratch.

+ Experiments on 2 tasks show that Scallop achieves similar or better performance compared to baselines with significantly less time.


## Questions and Weaknesses

- The submission missed a few closely related works. Both [A] and [B] improve the scalability of Neural Theorem Provers [28]. Similar to this paper, they notice that having to consider all proofs is the speed bottleneck, and they try to solve it by using only some proofs. Though their methods are different from Scallop. I think it would be helpful to have a short discussion of them in the related work section.  In addition, researchers have used provenance for synthesizing datalog programs ([C]), which is a different task but could also be worth mentioning.


[A] Minervini, Pasquale, et al. "Differentiable reasoning on large knowledge bases and natural language." Proceedings of the AAAI conference on artificial intelligence. Vol. 34. No. 04. 2020.
[B] Minervini, Pasquale, et al. "Learning reasoning strategies in end-to-end differentiable proving." International Conference on Machine Learning. PMLR, 2020.
[C] Raghothaman, Mukund, et al. "Provenance-guided synthesis of Datalog programs." Proceedings of the ACM on Programming Languages 4.POPL (2019): 1-27.

- Experiments in this paper do not present sufficient evidence that Scallop is flexible enough for real-world applications. On both tasks, there is a lot of task-specific prior knowledge built into the model. For example, the datalog programs are manually constructed. What if there is an elephant in the image but "elephant" is not in the vocabulary of the datalog program? I understand that some of these issues cannot be solved easily in one paper, but I would suggest at least stating these limitations clearly in the paper.

Experiments are narrow. Models have a lot of task-specific knowledge, e.g., manually constructed datalog programs. Compare with a baseline that sums the two numbers directly after classification.

- The paper lacks sufficient details about how gradients are computed in Eqn. 4. It looks like the authors are using dual numbers similarly to forward-mode AD. But it's not clear how dual numbers propagate through the inference procedure.

- The paper lacks sufficient details about the weighted model counting (WMC) procedure used for computing the score of the final answer. What is a WMC problem? Which WMC solver is used in the paper? Also, the paper says that scalability is limited by WMC solving (Line 78). I was surprised because I thought the limit was primarily in datalog solving.

- What is the datalog solver used in this paper? In the example in Fig. 1, how is the predicate "+" defined in datalog? Is it a built-in operator of the datalog solver?

- In Line 86, "a higher k ... accelerates the convergence of learning." Why?


## Comments on the Final Rating

I appreciate the idea of using only top-k proofs to speed up neuro-symbolic reasoning, and I think implementing it using provenance is a brilliant idea. However, the paper lacks details in many components, and the experiments are not as convincing as advertised in the intro. Therefore, I think the paper is not suitable for publication in its current form.

**Time Spent Reviewing:**

3

---

> ### Author Response · Authors · 2021-08-10
> **Response to Reviewer 2nJQ**
>
> Thank you for your insightful comments which we will incorporate into the revised version. We respond to your questions below.
>
> ### 1. Missing citations to GNTPs, CTP, and ProSynth:
>
> Thank you for pointing out these works.  We will discuss them in related work as follows:
>
> GNTPs applies numerical optimization to approximate the most likely proof among all possible proofs. This is similar to setting k=1 in Scallop.
>
> CTP introduces a neural network to select the rules to apply to the proof state in NTP. This is similar to using a neural network to select a subset of proofs rather than deterministically using top-k proofs in Scallop. Using such sampling-based methods to enhance Scallop’s performance is an interesting future research direction.
>
> ProSynth proposes a program synthesis algorithm that leverages provenance information from executing Datalog programs, similar to Scallop, although for a different task (ILP).  ProSynth computes provenance information in a post-processing step, after Datalog evaluation, and thus enumerates for each output fact a single proof without any particular property.  In contrast, Scallop computes provenance information during Datalog evaluation, and can thus track multiple proofs with a desired property (e.g. top-k) for each output fact.
>
> ### 2. Flexibility and limitations of Scallop’s application:
>
> We will clarify Scallop’s interface which requires the user to define the following:
>
> 1) a probabilistic relational database $D$ consisting of tuples with associated probabilities that encodes the output of the neural components, and
> 2) a Datalog program $P$ that specifies the logic reasoning components.
>
> We will also illustrate how the tasks in the paper, as well as additional tasks in the literature, can be expressed using this uniform interface (please see items **i** - **iv** in response to Reviewer NXqU).
>
> Finally, we will acknowledge the limitation of requiring the user to provide the Datalog program, and the need to automatically generate it by leveraging techniques in program synthesis, semantic parsing, and inductive logic programming.
>
> ### 3. What if the vocabulary of the program is insufficient to cover all real-life scenarios?
>
> Such issues can indeed arise due to the limitation of the knowledge base. One could address them using a more comprehensive knowledge base such as ConceptNet, which comprises 34M knowledge triplets, or even WikiData (comprising 94M knowledge triplets). As preliminary evidence, the current implementation of Scallop (provided in the supplementary material) is able to load a substantial subset of ConceptNet comprising 250K knowledge triplets and execute a query in 2 seconds. An alternative future direction is to use emerging language models (such as GPT-3/J) to construct knowledge triplets on demand. We will add this discussion in the revised version.
>
> ### 4. WMC procedure details: How are gradients calculated? Which WMC solver is used?
>
> The exact gradient calculation is explained in Section 5 of the DeepProbLog paper [1].  We use the same WMC solver used in the DeepProbLog implementation: the SDD package [2].
>
> ### 5. Scallop implementation details: What is the Datalog solver used? In the example in Fig. 1, how is the predicate "+" defined in Datalog?
>
> We implemented our own Datalog solver in Scallop, which comprises 25K lines of code in Rust.  The code is included as part of the supplementary material (under scallop folder), and we intend to make it open source upon publication. The predicate “+” is a built-in operator in our solver (along with other arithmetic and comparison operators).
>
> ### 6. In Line 86, "a higher k ... accelerates the convergence of learning." Why?
>
> When the resulting probability is computed, a higher k means more proofs will be considered. Subsequently, more gradients will be back-propagated to the source, resulting in a faster convergence of learning. During test time, on the other hand, k merely affects whether we consider certain low probability proofs. Therefore it will likely have less impact on the prediction result.
>
> ### 7. Reference
>
> [1]  RAEDT, L. D. et. al. DeepProbLog: Neural probabilistic logic programming. NeurIPS (2018).
>
> [2] Choi, Arthur, and Adnan Darwiche. The SDD Package, 8 Jan. 2018, reasoning.cs.ucla.edu/sdd/.

---

> > ### Comment · Reviewer_2nJQ · 2021-09-11
> > **Final Comments**
> >
> > Hi,
> >
> > I have read the rebuttal, and there were some discussions among the reviewers. The rebuttal addresses some of my concerns. In particular, I think the datalog solver the authors developed could also be an interesting technical contribution. I'm not aware of existing datalog solvers whose provenance computation can be configured to perform the top-k calculation needed in this paper.
> >
> > At this moment, my main concern is that the proposed system is fairly narrow and requires a lot of domain-specific manual designs. This limitation is kind of expected since it is challenging to build a general and flexible symbolic system without manual design. But I think the authors failed to articulate this limitation in writing. As a result, it feels like the intro oversells what has been really achieved.
> >
> > I'm on the fence with this submission. I do not plan to change my score, but I'm fine with either accept or reject.
> >
> > Best,

---

> > > ### Author Response · Authors · 2021-09-14
> > > **Thank you**
> > >
> > > Thank you for your comment. It was not our intention to oversell our contributions, and this exchange is helpful to us for improving the presentation, especially in terms of clarifying the strengths and limitations of our approach in the introuction, which we will do in the revised version.

---

### Official Review · Reviewer_uqGA · 2021-07-16

**Rating:** 6
**Confidence:** 1

**Summary:**

The authors introduce a framework that combines neural modules for visual perception with a probabilistic model that computes the top-k proofs of objects being in images given a logical rule.

**Limitations And Societal Impact:**

Limitations and societal impact were not discussed. I do not believe the method has any negative societal impact.

**Main Review:**

I find it difficult to evaluate this paper. While some literature exists with some similar approaches, [1] there is no direct comparison with other probabilistic methods that compute proofs and only a comparison with one method that uses knowledge graphs. While the authors use LXMERT to compare, they do this on their own task that they develop as an extension from GQA but they use a Recall@5 metric which makes the LXMERT results not interpretable to previous results on GQA.

**Originality**: It is not new to use some form of truncated proof evaluation to make methods like this computationally feasible. While top-k truncation[2] and learned truncation[1] have been employed to shorten the paths of proofs, I believe, as I understand the paper, that the authors include the top-k proofs in downstream computation to reduce the computational burden. To me, this seems to be a novel contribution -- albeit I am not sure this is true.

**Quality**: The quality of evidence that the authors provide for their method is poor. It is difficult to trust a new dataset that was designed for the author's method where the report results are a non-standard metric. The results would be much more trustworthy if they would be more closely tied to the existing literature (either in VQA, or some kind of neural theorem provers).

**Clarity**: The paper was easy to read and most details were clear to me.

**Significance**: The results vs LXMERT look impressive but I also find that it is difficult to interpret these results. As such, it is difficult to say how significant these results are truly are.

[1] Learning Reasoning Strategies in End-to-End Differentiable Proving
[2] End-to-End Differentiable Proving

**Time Spent Reviewing:**

1

---

> ### Author Response · Authors · 2021-08-10
> **Response to Reviewer uqGA**
>
> Thank you for your comments and questions.
>
> ### 1. Customized VQA dataset and choice of Recall@5 metric.
>
> We created VQAR because we wish to evaluate the effectiveness of logic reasoning in solving tasks that involve *multi-hop* reasoning in addition to perception. The original visual question and answering (VQA) tasks only focus on the perception component but not the reasoning ability.  Our dataset not only covers pure visual questions, but also includes questions that require consulting an external knowledge base.
>
> The original VQA task can be expressed in VQAR with a slight modification. For example:
> -  **VQA**
>    * Question: Does there exist a tall animal on the left?
>    * Answer: Yes.
> - **VQAR**
>    * Question: Find the tall animals on the left.
>    * Answer: object 2 and object 12.
>
> Notice that the VQAR task demands more information than the VQA task, and thus evaluating its accuracy involves performing a set to set comparison.  We therefore adopt a standard metric Recall@5 as the evaluation metric, instead of match or not.
>
> As discussed in Appendix B.1, we considered alternative datasets to perform multihop reasoning, such as KB-VQA and OK-VQA. However, they incorporate large noisy knowledge bases, and do not provide fine-grained control over the question difficulty.
>
> Lastly, while the VQAR task involved customization for the reasons explained above, we note that the mathematical and logical reasoning tasks in our evaluation are identical to those in prior work on DeepProbLog.
>
> ### 2. Comparison to works on NTP and CTP:
>
> CTP introduces a neural network to select the rules to apply to the proof state in NTP. This is similar to using a neural network to select a subset of proofs rather than deterministically using top-k proofs in Scallop. Using such sampling-based methods to enhance Scallop’s performance is an interesting future research direction.  We thank the reviewer for mentioning this paper and will incorporate this discussion in related work.

---

> > ### Comment · Reviewer_uqGA · 2021-09-01
> > **Thank you**
> >
> > I can see the difference between soft and hard selection of subsets of proofs. If you add a discussion I am happy to up my score by 1.
> >
> > Furthermore, the discussion of the other reviewers convinces me that this research area is underexplored and I would be happy to give this paper a bit more exposure. I think with this reasoning I lean slightly towards acceptance and will give the paper of score 6.

---

> > > ### Author Response · Authors · 2021-09-02
> > > **Discussion of soft and hard proof selection**
> > >
> > > Thank you for your feedback. We are glad that we, along with other reviewers, helped to address your concerns. We will incorporate a discussion of the difference between the soft and hard selection of proofs in the related work section:
> > >
> > > NTP, GNTP, and CTP operate top-down and dynamically construct neural modules in proof states to estimate the success score of the proofs. Since NTP suffers from high computational complexity, GNTP and CTP employ various techniques to select subsets of proof states. GNTP approximates the most likely proofs in each proof state using nearest neighbor search. CTP applies soft proof selection by training a neural network to select the rules, deriving proofs individually. On the contrary, Scallop executes the program symbolically and keeps provenance information. It uses top-k to deterministically select the most relevant proofs during bottom-up logical deduction. This hard proof selection process suppresses a large number of spurious proofs while keeping the most related ones. However, using a soft proof selection in Scallop can potentially help accelerate convergence and is an interesting future research direction. Lastly, we note that NTP does not support recursion (applying the same rule more than once).

---

### Official Review · Reviewer_34Ja · 2021-07-16

**Rating:** 7
**Confidence:** 2

**Summary:**

This paper presents a method for scaling probabilistic reasoning by using only the top-k proofs in computing the probability of each outcome. Their algorithm, Scallop, allows tuning k at both training and test time, allowing for larger k at training and smaller k at test (and therefore speedier inferences). Empirically, Scallop outperforms leading, contemporary methods on a challenging VQA task requiring multi-hop reasoning. Further, Scallop shows clear speed ups in training wall clock time with respect to typical neural logic programming approaches that do not perform Scallop's k-width beam search over proofs.

**Limitations And Societal Impact:**

- The authors should address the issues of bias that can result in relying on automated decisions from a probabilistic database, and that such systems ultimately rely on relational rules that can be highly biased.

**Main Review:**

This paper introduces a straightforward idea for speeding up the inference time of differentiable probabilistic reasoning systems. The paper presents a clear formalism and theoretical justification for soundness of their core technique of a k-width beam search over proofs. The experimental section shows clear gains in wall-clock time for both training and inference compared to baselines, as well as Scallop's ability to generalize well to more challenging VQAR questions. Overall, I find the paper makes a compelling case for the benefits of this method.

A few suggestions for improving understanding of this method:

- It would be revealing to look at which kinds of queries lead to incorrect outputs by Scallop, and whether these errors are due to errors in the differentiable layers or due to limiting the proof budget to *k*. The proportion of errors due to the latter is the real error rate that matters in comparison to baselines making use of the full set of proofs.
- Of course, it would be useful to know how sensitive Scallop is to various hyperparameters, and importantly, how learning rate and batch size might impact the best choice of k.

**Time Spent Reviewing:**

4

---

> ### Author Response · Authors · 2021-08-10
> **Response to Reviewer 34Ja**
>
> Thank you for your appreciation and useful feedback which will improve the presentation of our work.
>
> ### 1.  Which kinds of queries lead to incorrect outputs by Scallop? Are these due to errors in the differentiable layers or due to limiting the proof budget to k?
>
> We will incorporate a qualitative analysis of the errors. ​​Scallop’s formal guarantee (Proposition A.11) ensures that the output probability order is the same across all different choices of k, under the assumption that no facts involved in “annotated disjunctions” occur in the same proof.  If this assumption is violated, however, we may introduce errors in the final output (unless a high enough k is provided). Fortunately, this occurs rarely in the VQAR dataset. We inspected all of the 1,436 mispredicted samples in the test dataset of the VQAR benchmark using k=10, and were able to find only 36 such cases; further, 27 out of the 36 cases can be correctly predicted with k=50. Therefore, we believe that in the vast majority of cases, the errors happen in the differentiable layers.
>
> ### 2. How sensitive is Scallop to various hyperparameters, and importantly, how does learning rate and batch size impact the best choice of k?
>
> We will clarify how the user can tune the hyperparameter k in practice. We found k=5 to be a suitable default value to balance accuracy and training cost for both the tasks in our paper.  One can increase or decrease this default value in order to achieve higher accuracy or lower training cost, respectively.
>
> Empirically, we were not able to observe a noticeable impact of other learning parameters on the best choice of k. But we acknowledge the reviewer’s observation that, in principle, k is not independent of those other parameters. Intuitively, batch size and learning rate determines the gradient values in the backpropagation process. On the other hand, our parameter k affects the loss value, and thereby also impacts the gradient values.  Therefore, they are not independent in principle.

---

> > ### Comment · Reviewer_34Ja · 2021-08-17
> > **Thanks for the responses**
> >
> > Thanks for addressing my concerns. After reading the other reviews and realizing the missing discussion around scalability along alternative axes and the missing discussion around NTP/CTP, I decided to lower my score to a 7. I still believe this to be a solid contribution, but now see that there are a few missing details—mentioned in the other reviews—whose inclusion would benefit this paper.

---

### Official Review · Reviewer_NXqU · 2021-07-19

**Rating:** 6
**Confidence:** 3

**Summary:**

This work presents a combination of techniques for learning and probabilistic reasoning over databases to improve visual question answering (and reasoning) performance. A perception model reads the visual input and converts it into an interpretable format (which can be, e.g., a probabilistic database or a scene graph) that is then used for probabilistic reasoning together with a (fixed) knowledge graph. This reasoning produces the top-k explanations for a result.

**Main Review:**

Pros:
+ Interesting approach and combination of techniques: The paper combines probabilistic reasoning over knowledge graphs, datalog, weighted model counting, and deep learning (for the perception parts) in a meaningful way. On the technical level this is quite an achievement.
+ Significant gains on benchmarks that involve image recognition and simple reasoning steps.
+ Reasonable scalability: costly algorithms, such as weighted model counting, are applied in a way that does not immediately limit the scalability.

Cons:
- Complexity of the approach: The approach combines knowledge graphs, datalog, weighted model counting and backprop. This means that there are many moving parts and it is hard for me to understand how the system could be applied to a new task.
- Significant parts of the architecture have to be adapted to each task (scene graphs, logic programs matching the task).
- toy benchmark/custom benchmark; claims to be real-world task, but I'm not totally convinced it is
- The paper highlights the introduction of a tunable parameter as a contribution, which I find confusing: I prefer methods that have fewer (hyper)parameters that I have to tune.

The writing style is okay, but many parts could be written in a more concrete way. Some examples:
- Many claims are very general and abstract, and while reading the paper I often missed concrete evidence for claims. E.g. the first two sentences of the abstract.
- The introduction would benefit from clarifications on what the terms "proof", "provenance", and "reasoning granularity" mean exactly in the context of this work.
- The introduction promises an evaluation on a "real-world task", but does not go into what this task is.

Questions to the authors:
- I do not fully understand the VQAR task and believe some crucial details are missing: what exactly are the inputs and what does your model predict? In particular, I wonder about the scene graph. In the introduction you write that the scene graph has to be predicted by the vision model; how exactly does this work?
- How did you split the data into train/test for the VQAR task? Do the queries from the train and test set share the same pictures?
- Scalability: The paper claims to present a scalable method, but there are no experiments on how well it scales with the size of the knowledge graph, the queries, or images. Do I understand correctly that the main claim is that the method scales better than DPL? Could you discuss the limits of scalability?
- The examples in Figure 1 look like you just need DNF counting instead of WMC counting? Why do you need WMC?

Nitpicks:
- line 78, 79: "WMC is at least #P-complete" -> should be either "is at least #P hard" or "is #P complete". Also, your approach does not need to solve WMC exactly, but approximate solutions are sufficient. Then the problem is not #P hard anymore. See works by Kuldeep Meel. However, this does not invalidate the underlying point that the authors want to make: WMC is a hard problem.
- Check the capitalization of acronyms in the references ...

**Time Spent Reviewing:**

3

---

> ### Author Response · Authors · 2021-08-10
> **Response to Reviewer NXqU**
>
> Thank you for your insightful comments which we will incorporate into the revised version. We respond to your questions below.
>
> ### 1. Generality of approach: system has many moving parts, how to generalize it to other tasks, significant parts have to be adapted to each task.
>
> We will clarify the distinction between Scallop’s ***implementation***, which indeed has multiple components, and the ***interface***, which encapsulates these components and requires the user to only define the following:
>
> 1) a probabilistic relational database $D$ consisting of tuples with associated probabilities that encodes the output of the neural components, and
> 2) a Datalog program $P$ that specifies the logic reasoning components.
>
> To demonstrate generality, we illustrate how this interface can uniformly capture a variety of different tasks that involve perception and reasoning, including the two tasks from the paper (**I** and **ii** below) and additional tasks from the literature (**iii** and **iv** below):
>
>
> **i)** For the synthetic tasks in our paper, $D$ represents the output of the MNIST digit recognition network as tuples of the form digit(IMAGE_ID,NUMBER), and $P$ represents the logic rules for addition/sorting.
>
> **ii)** For the VQAR task in our paper, $D$ represents the output of the three MLP classifiers, producing three kinds of relations "name(o1, giraffe)", "attr(o1, tall)", and "rela(left, o1, o2)", and $P$ contains the full knowledge base (3K knowledge triplets) and the programmatic query.
>
> **iii)** Formula parsing and evaluation from [1]: In this task, a vision model takes an image of a hand-written formula (e.g. "3 * (1 + 2)"), and $D$ encodes its output using probabilistic relations of the form "constant([3], 3)" and "binary_op([*], ‘*’, [3], [(1 + 2)])" (where [x] denotes the id of a bounding box in the image that contains x).  The Datalog program $P$ contains rules for formula evaluation, such as “eval(F, LY + RY) :- binary_op(F, ‘+’, L, R), eval(L, LY), eval(R, RY).”
>
> **iv)** Natural language reading comprehension from [2], [3]: In this task, a language model takes as input a natural language sentence (e.g. "tom kicks the ball") and $D$ encodes its output using probabilistic relations of the form "subject_verb(tom, kicks)”, “verb_object(kicks, ball)", and "event(kicks, tom, ball)". A natural language query “who kicks the ball?” can be written as a programmatic query “target(W) :- event(kicks, W, ball)”, represented using the Datalog program $P$.
>
> In summary, we expect Scallop to be used as a general probabilistic reasoning framework. An interesting but orthogonal future direction concerns automatically generating the program $P$ using techniques from program synthesis, semantic parsing, and inductive logical programming.
>
> We will revise Section 4 with the above discussion elucidating Scallop’s interface and its instantiations for different tasks.
>
> ### 2. Description of VQAR benchmark is unclear:
>
> We will improve the presentation by addressing your questions as follows:
>
> - Dataset splits: Queries from the train and test set *do not* share the same pictures.​ This is currently stated in Appendix B but we will clarify it in the main body of the paper:
>
>   + “For each image, we generate 10 different question and answer pairs for each clause length 2 to 6, to obtain 4 million data points in total. We split them into training (60%) validation (10%), and testing (30%) sets, and ensure that all the questions about the same image occur within the same split to test generalizability.”
>
> - Clarification of scene graph: As noted in the illustration of Scallop’s user interface above, the scene graph refers to the probabilistic relational database comprising three relations: “name”, “attr”, and “rela”. The scene graph is computed as follows:
>
>    + A pre-trained Fast-RCNN network takes in the raw image and produces object bounding boxes and feature vectors, which in turn are fed to three MLP classifiers, M_name, M_attr, and M_rela, to predict the tuples in these three relations simultaneously. For example, the M_name classifier will take in o1’s bounding box and its feature vector, and produce a distribution of the classified names: “0.81::name(o1, tiger); 0.15::name(o1, giraffe); ...”. On the other hand, M_rela takes in two bounding boxes and feature vectors from, say, ox and oy. It will then produce a distribution of classified relations between ox and oy: “0.15::rela(‘on’, ox, oy). 0.05::rela(‘behind’, ox, oy). ...”
>    + Further details about the model are provided in Appendix C.2.
>
>
> ### 3. Scalability claim and limits are unclear:
>
> As the reviewer noted, our scalability claim is only with respect to DPL. Scallop scales significantly better than DPL, with a modest loss of accuracy in practice.
>
> As the reviewer also noted, one can evaluate this claim with respect to the size/complexity of different components in the VQAR task: image, query, knowledge base. We discuss each of these metrics below.
>
> 1. Image.  We employ real-world images from Visual Genome, which contain a rich variety of 500 names, 609 attributes and 229 relationships, as opposed to synthetic images as in CLEVR (which uses only 15 attributes and 4 relationships).
>
> 2. Query.  We generate programmatic queries of varying complexity, from C2 to C6, which mimics the complexity of natural language queries occurring in practice (e.g. the average size of a programmatic query in the GQA dataset is around C3-C4).  As Table 2 shows, our accuracy remains almost the same for queries of different complexity.
>
> 3. Knowledge base.  We did not evaluate scalability with respect to this metric.  Indeed, as the reviewer notes, it impacts the time per query (which is 0.2 seconds on average for our current knowledge base of 3K triplets) which in turn impacts the time to train the combined model (which took 3 days).  The time per query increased to 2 seconds when we use a subset of the ConceptNet knowledge base comprising 250K triplets (using the implementation from the supplementary material).  While promising, however, it increases training time to 30 days.  We have not evaluated Scallop on even larger knowledge bases such as the entire ConceptNet (34M) or WikiData (94M). Besides improving Scallop’s scalability on larger knowledge bases, an interesting future direction is to use emerging language models (such as GPT-3/J) to construct knowledge triplets on demand. We will add this discussion in the revised version.
>
> ### 4. DNF counting v.s. WMC
>
> We thank the reviewer for pointing out weighted DNF counting and will discuss it in the revised version. It is unclear to us how to perform weighted DNF counting using the off-the-shelf WMC solver in Scallop’s implementation.  More generally, we are extending the Scallop framework to support richer forms of reasoning such as negation and aggregation, which will necessitate WMC. Nevertheless, we acknowledge this important optimization possibility, which could be incorporated into the WMC solver to further improve the overall efficiency of Scallop on tasks for which weighted DNF counting is sufficient.
>
> ### 5. Hyperparameter tuning
>
> We will clarify how the user can tune the hyperparameter k in practice.  We found k=5 to be a suitable default value that balances accuracy and training cost for both tasks presented in our paper. One can increase or decrease this default value in order to achieve higher accuracy or lower training cost, respectively.
>
> ### 6. Reference
> [1]: Zhu, Song-Chun et al. “Closed Loop Neural-Symbolic Learning via Integrating Neural Perception, Grammar Parsing, and Symbolic Reasoning.” ICML (2020).
>
> [2]: Ball, Spencer et al. “CUAD: An Expert-Annotated NLP Dataset for Legal Contract Review”. Arxiv Preprint (2021).
>
> [3]: Dua, Dheeru et al. “DROP: A Reading Comprehension Benchmark Requiring Discrete Reasoning Over Paragraphs.” NAACL (2019).

---

### Author Response · Authors · 2021-09-01
**Further Discussion and Comment?**

First of all, we thank all the reviewers for your insightful feedback. As the discussion period is coming to an end, we wonder if there is any suggestion, concern, or unclarity that we can further address before your conclusion. Please don’t hesitate to let us know and we are more than happy to continue the conversation.

---

### Decision · Program_Chairs · 2021-09-27

**Decision:**

Accept (Poster)

**Comment:**

Four knowledgeable reviewers praised the idea of combining forward chaining with differentiable and probabilistic programming to tackle tasks such as VQA where the combination of symbolic and sub-symbolic systems clearly pays off.

During the rebuttal, authors provided enough context and details to better cast the proposed Scallop into the literature and to better evaluate the experiments.

The paper is accepted, as it can spawn further discussion in the neuro-symbolic community about such an important research direction.
However, acceptance is conditional on the authors including all the promised (and discussed) details in the camera-ready. One additional point to clarify in the camera-ready is how Scallop differs from modern implementation of (Deep) ProbLog where essentially forward reasoning is used.